# Status of Routine Immunization Coverage in the World Health Organization African Region Three Years into the COVID-19 Pandemic

**DOI:** 10.3390/vaccines12020168

**Published:** 2024-02-07

**Authors:** Franck Mboussou, Sarah Kada, Maria Carolina Danovaro-Holliday, Bridget Farham, Marta Gacic-Dobo, Jessica C. Shearer, Ado Bwaka, Adidja Amani, Roland Ngom, Yolande Vuo-Masembe, Charles Shey Wiysonge, Benido Impouma

**Affiliations:** 1World Health Organization, Regional Office for Africa, Brazzaville P.O. Box 06, Congo; 2Momentum, JSI, Arlington, VA 22202, USA; 3World Health Organization Headquarters, Avenue Appia 20, 1211 Geneva, Switzerland; 4PATH, 455 Massachusetts Avenue NW, Washington, DC 20001, USA; jshearer@path.org

**Keywords:** routine immunization, vaccination coverage, zero-dose, WUENIC, COVID-19, catch-up, African Region

## Abstract

Data from the WHO and UNICEF Estimates of National Immunization Coverage (WUENIC) 2022 revision were analyzed to assess the status of routine immunization in the WHO African Region disrupted by the COVID-19 pandemic. In 2022, coverage for the first and third doses of the diphtheria–tetanus–pertussis-containing vaccine (DTP1 and DTP3, respectively) and the first dose of the measles-containing vaccine (MCV1) in the region was estimated at 80%, 72% and 69%, respectively (all below the 2019 level). Only 13 of the 47 countries (28%) achieved the global target coverage of 90% or above with DTP3 in 2022. From 2019 to 2022, 28.7 million zero-dose children were recorded (19.0% of the target population). Ten countries in the region accounted for 80.3% of all zero-dose children, including the four most populated countries. Reported administrative coverage greater than WUENIC-reported coverage was found in 19 countries, highlighting routine immunization data quality issues. The WHO African Region has not yet recovered from COVID-19 disruptions to routine immunization. It is critical for governments to ensure that processes are in place to prioritize investments for restoring immunization services, catching up on the vaccination of zero-dose and under-vaccinated children and improving data quality.

## 1. Introduction

The COVID-19 pandemic substantially disrupted routine immunization services in the World Health Organization (WHO) African Region, leading to a precipitous decline in childhood vaccination rates compared to pre-pandemic levels [1,2,3,4]. The African Region is one of six WHO regions and comprises 47 of the 54 countries on the African continent [5]. The first laboratory-confirmed case of COVID-19 in the African Region was reported on 25 February 2020 in Algeria [6]. COVID-19 then quickly spread to all countries in the African Region. In addition to setting up surveillance and case-management systems, governments in the region implemented non-pharmaceutical interventions to minimize social contacts, including but not limited to partial or full lockdowns [7]. The COVID-19 lockdowns have had adverse consequences on health service provision and utilization. Childhood routine immunization is among the public health services disrupted by response measures to the COVID-19 pandemic, leading to parents avoiding routine immunization sessions due to fear of becoming infected with COVID-19 [8], the postponement of mass vaccination campaigns and the diversion of resources due to the COVID-19 pandemic response in an already underfunded health system [9]. Accordingly, the COVID-19 pandemic significantly contributed to a decline in routine immunization coverage, resulting in a greater number of zero-dose children (i.e., children who have not received any routine vaccination and, more specifically, missed the first dose of the diphtheria–tetanus–pertussis-containing vaccine—DTP) and under-immunized children (i.e., children who missed all due vaccines as per the national immunization schedule within the first year of age and, more specifically, are missing the third dose of the DTP vaccine [10]).

To speed up the recovery of immunization services, Immunization Agenda 2030 (IA2030) partners, including the WHO, UNICEF and Ministries of Health, joined forces in April 2023 to call for “The Big Catch-Up”, a global push to increasing vaccination coverage among children to pre-pandemic levels or above, with a particular focus on countries that recorded the highest number of zero-dose children [10]. This Big Catch-Up is designed to help countries catch children up on vaccines they missed during the pandemic, recover from COVID-19’s impact on routine immunization and strengthen health systems to better integrate immunizations into primary health care with the view of achieving the IA2030 trajectory [10]. The IA2030 is an ambitious global strategy to maximize the lifesaving impact of vaccines which, if fully implemented, will save 50 million lives by 2030. It seeks to create a world in which everyone, everywhere, at every age fully benefits from vaccines for good health and well-being [11].

Given the lifting of the public health emergency of international concern (PHEIC) status for the COVID-19 pandemic by the WHO on 5 May 2023 [12], it is critical to assess the performance of routine immunization services in the African Region. This paper summarizes the trend and geographical distribution of immunization coverage in the WHO African Region as well as number of zero-dose and under-immunized children in the Region from 2019, representing the pre-pandemic period, to 2022, representing the end point of the emergency phase of the COVID-19 pandemic.

## 2. Materials and Methods

A retrospective descriptive analysis of secondary data related to routine immunization in the WHO African Region at the end of 2022 compared to the pre-COVID-19 pandemic level (2019) was conducted.

### 2.1. Inclusion and Exclusion Criteria

Countries in the WHO African Region that reported data on routine immunization coverage to the WHO and UNICEF in 2022 through Joint Reporting Forms (JRFs) [13] were included. All 47 countries in the WHO African Region met this criterion and were included. The seven African countries that are part of the WHO Eastern Mediterranean Region (Djibouti, Egypt, Libya, Morocco, Somalia, Sudan and Tunisia) were excluded.

### 2.2. Data Sources and Measurement

The 2022 revision of the WHO and UNICEF Estimates of National Immunization Coverage (WUENIC) and administrative data were used for this analysis. Administrative data are collected by health facilities and reported by each country to the WHO and UNICEF through JRFs [13]. The WHO and UNICEF use historic time series and other available sources of information to build and adjust immunization coverage, providing WUENIC data [14]. WHO and UNICEF estimates are country-specific and limited to the national level.

The primary indicator (variable) of interest in this study was the national-level coverage of each antigen or combination of antigens. Coverage was measured as the number of children immunized with each antigen/combination of antigens divided by the target population.

Fourteen antigens or combinations of antigens were tracked in the 2022 revision of the WUENIC (released in July 2023) [15]. All the following 14 antigens or combinations of antigens were considered in this study: Bacille Calmette–Guérin (BCG), measles-containing vaccine (first dose—MCV1; second dose—MCV2)), diphtheria–tetanus–pertussis-containing vaccine (first dose—DTP1; third dose—DPT3), oral poliovirus vaccine (third dose—POL3), inactivated poliovirus vaccine (first dose—IPV1), the first dose of the rubella-containing vaccine (RCV1), hepatitis B vaccine (birth dose—HEPBB; third dose—HEPB3), the third dose of the *haemophilus influenzae* type b vaccine (HIB3), the yellow fever vaccine (YFV), rotavirus vaccine (last dose—ROTAC) and the pneumococcal conjugate vaccine (third dose—PCV3). HEPBB coverage estimates were considered only for countries with the ability to record data within 48 hours of delivery. Some countries provided data on children vaccinated with the HEPBB but were unable to record vaccinations in a timely manner; therefore, coverage estimates were not made.

### 2.3. Data Analysis

Data from the WUENIC from 2015 to 2022 [15] were analyzed, and pre-pandemic (2019) and 2022 immunization coverages were compared.

In addition to the immunization coverage for each antigen or combination of antigens (provided by the WUENIC 2022 revision), the following parameters were calculated:The percentage of difference in immunization coverage in 2022 compared to 2019 (pre-pandemic period): the coverage of the year under review divided by the 2019 coverage minus one and multiplied by one hundred. To calculate a percentage difference in the few countries in which coverage was null in 2019 but non-zero in 2022, coverage in 2019 was set to one.The number of zero-dose (ZD) children: the number of surviving infants (aged 0–11 months) for a given year (from the United Nations (UN) Population Estimates 2022 revision) [16] minus the number of children vaccinated with the DTP1 during the same year.The number of un-immunized children for vaccines other than the DTP: the number of the target population for a given year and antigen (from UN Population Estimates 2022 revision) [16] minus the number of children vaccinated with the related vaccine during the same year.The number of under-immunized children: the number of children vaccinated with the last dose of a given vaccine minus the number of children vaccinated with the first dose of the same vaccine.The percentage of ZD children: the number of zero-dose children divided by the number of surviving infants for the same period, multiplied by 100.The number ZD children per one thousand population: number of zero-dose children divided by the total population (UN population Estimates) for the same period, multiplied by 1000.

Countries’ administrative coverages in 2022 for DTP3 were compared to WHO and UNICEF estimates. Administrative coverages are self-reported by national authorities to the WHO and UNICEF on a yearly basis through JRFs and are calculated using the number of doses administered from health facilities as the numerator and national estimates of target populations as the denominator.

All data analyses and visualizations were performed using R software, version 4.1.2 [17].

## 3. Results

### 3.1. Overview of Data Reported

Data on immunization coverage in 2022 were available for 47 countries for the BCG, DTP1, DTP3, HEPB3, HIB3, MCV1 and POL3. The number of countries with data available was 41 for the MCV2, 40 for the PCV3, 38 for the ROTAC, 32 for the RCV1, 24 for the YFV and 11 for the HEPBB. Countries that did not report on these antigens (MCV2, ROTAC, RCV1, YFV and HEPBB) are those that have not yet introduced them in their national schedules.

### 3.2. Immunization Coverage Trends

The number of surviving infants vaccinated with the DTP1 in 2022 was 30.8 million children versus 30.4 million in 2021, providing nearly 350,000 additional children immunized. The percentages of children having received the DTP1 and DTP3 in 2022 were estimated at 80% and 72%, respectively. DTP1 coverage decreased from 83% in 2019 to 81% in 2020 before plateauing at 80% in 2021 and 2022, while DTP3 coverage declined from 77% in 2019 to 74% in 2020 before plateauing at 72% in 2021 and 2022. Figure 1 presents the distribution of yearly immunization coverages with the 14 antigens/combinations of antigens analyzed in this study from 2015 to 2022.

The number of children vaccinated with the MCV1 in 2022 was 26.5 million children compared to 26 million in 2021: that is nearly 484,000 additional children immunized. MCV1 coverage in 2022 in the WHO African Region was estimated at 69%, moving from 71% in 2019 to 70% in 2020 before fluctuating between 68% in 2021 and 69% in 2022. The same trends were observed with BCG, HEPB3, HIB3, IPV1, POL3, PCV3, RCV1, ROTAC and YFV.

An upward trend between 2019 and 2022 was observed with the HEPBB and MCV2. Fifteen countries have introduced the HEPBB in their schedule, including four between 2015 and 2019 and three between 2020 and 2022. Forty countries have introduced the MCV2 in their schedule including 18 before 2015, 12 between 2015 and 2019 and 10 between 2020 and 2022.

Figure 2 presents the immunization the distribution of coverage in 2022 for the DTP1, DTP3 and MCV1 with countries grouped by sub-region. The median coverages for the DTP1, DTP3 and MCV1 were 89% [range: 53%; 99%], 82% [range: 42%; 99%] and 79% [range: 37%; 98%], respectively. The WHO target of 90% or greater coverage for the DTP3 was achieved in 2022 in 13 out of 47 countries (28%).

Figure 3 presents a country-specific comparison of immunization coverage in 2022 and in the pre-pandemic period (2019) for the 14 antigens or combinations considered in this study. The immunization coverage in 2022 compared to 2019 decreased by more than 10% in 4 countries (8%) for the DTP1, in 7 countries (15%) for the DTP3 and in 14 countries (25%) for the MCV1. The number of countries that recorded an increase by at least 10% in immunization coverage in 2022 compared to 2019 was three (6%) for the DTP1, four (8%) for the DTP3 and six (13%) for the MCV1. The highest proportions of countries for which immunization coverage increased by at least 10% in 2022 compared to 2019 were recorded for the HEPBB (5 countries out of 11 that reported to the WHO and UNICEF, 71%), MCV2 (17 countries out of 41 reporting, 41%) and ROTAC (6 countries out of 38 reporting, 16%). This may be due to the fact that few countries introduced these vaccines during the pandemic period: three countries introduced the HEPBB, ten countries introduced the MCV2 and one country introduced the rotavirus vaccine. Three countries recorded the highest proportions of antigens/combination of antigens with an over 10% increase in immunization coverage in 2022 compared to 2019: Chad (10 antigens/combinations of antigens out of 14, 71%), Liberia (7 antigens/combinations of antigens, 50%) and South Sudan (7 antigens/combinations of antigens, 50%). In these three countries, immunization coverages for the three main antigens or combinations of antigens (DTP1, DTP3 and MCV1 were low in 2019 and increased in 2022 but remained low except for the DTP1 in Liberia. In Chad, the coverage increased from 65% to 74% with the DTP1, from 50% to 60% with the DTP3 and from 40% to 56% with the MCV1; in Liberia, the coverage increased from 86% to 93% with the DTP1, from 70% to 78% with the DTP3 and from 68% to 79% with the MCV1; in South Sudan, the coverage increased from 68% to 76% with the DTP1, from 61% to 73% with the DTP3 and from 65% to 72% with the MCV1.

### 3.3. Un- and Under-Immunized Children

In 2022, the WHO African Region recorded 7.7 million zero-dose children (20% of surviving children) compared to 7.6 million (20% of surviving children) in 2021, 7.1 million (19% of surviving children) in 2020 and 6.2 million (17% of surviving children) in 2019. Cumulatively, the number of zero-dose children from 2019 to 2022 in the WHO African Region was estimated at 28.7 million, representing 19.0% of the four cohorts of surviving infants (Table 1).

Figure 4 presents the geographical distribution of the number of zero-dose children as well as the number of zero-dose children per one thousand population in the WHO African Region.

Nigeria, Ethiopia, the Democratic Republic of the Congo, Angola, the United Republic of Tanzania, Madagascar, Mozambique, Mali, Chad and Cameroon were the top 10 countries by number of zero-dose children recorded from 2019 to 2022, accounting for 80.3% of the total number of zero-dose children in the region. Nigeria, Ethiopia, the Democratic Republic of the Congo and the United Republic of Tanzania, the four most populated countries, accounted for 57.7% of the total number of zero-dose children. 

The number of surviving children who missed the first dose of MCV was estimated at 12.1 million (31% of surviving children) in 2022 versus 10.8 million (29%) in 2019. Figure 5 presents the distribution of the number of children not immunized with the DTP (zero-dose children) and MCV in the top 10 countries by the number of zero-dose children.

Figure 6 presents the percentage of children not immunized with DTP and MCV in 2019 and 2022. The percentages of zero-dose children and children not immunized with the MCV increased in 2022 compared to 2019 in 24 countries (51%) and 27 countries (57%), respectively. Mozambique (+24%), Angola (+14%), Gabon (+12%) and Madagascar (+11%) experienced the most dramatic increase in zero-dose children (Figure 6A), while Sao Tome and Principe (+18%), Madagascar (+16%), Angola (+14%), Niger (+14%), Gambia (+11%), Gabon (+10%) and Malawi (+10%) recorded the largest increase in children not immunized with the MCV (Figure 6B). Three countries recorded the highest decline in the proportion of children not immunized with the MCV in 2022 compared to 2019: Chad (−15%), Liberia (−11%) and Namibia (−11%; Figure 6B).

### 3.4. WUENIC and Administrative Coverage Data Comparison

The DTP3 was used as tracer for comparing administrative (i.e., reported by countries) and WUENIC coverages in 2022. Sixteen countries (47%) recorded a percentage difference between −10% and 10%, while 19 countries (40%) recorded more than a 10% difference (indicating that the WUENIC coverage is less than the administrative coverage) and six countries (13%) with below a −10% difference (indicating that the administrative coverage is less than the WUENIC coverage; Figure 7). The following six countries reported less administrative coverages than the WUENIC coverage: Botswana, Comoros, Eritrea, Mauritius, Sao Tome and Principe and Senegal. Finally, six countries (Benin, Cape Verde, Ghana, Niger, Rwanda and the United Republic of Tanzania) reported greater than 100% DTP3 administrative coverage.

## 4. Discussion

The COVID-19 pandemic has contributed to the disruption of routine vaccination services in the WHO African Region, resulting in decreased immunization coverage for lifesaving vaccines and increased vulnerability to vaccine-preventable diseases for millions of children [10,18,19]. This study, using the latest WUENIC data at the time of publication, shows that immunization coverage for most routine vaccines in the WHO African Region, in 2022, has not yet reached pre-pandemic levels. Overall coverage with the first and third doses of the DTP vaccine plateaued at 80% and 72% in 2021 and 2022, respectively, after a sharp decline in 2020. Only 13 countries out of 47 (28%) achieved the global target coverage of 90% or above with the DTP3 in 2022. Worldwide, DTP1 coverage increased from 86% in 2021 to 89% in 2022 but remained below the 90% coverage achieved in 2019 [20,21]. Similarly, DTP3 coverage increased globally from 81% in 2021 to 84% in 2022 but remained below the 2019 level (86%). Unlike the WHO African Region, DTP3 coverage has recovered to pre-pandemic levels in the South-East Asian, Eastern Mediterranean and the American WHO Regions [20,22].

Conflicting public health priorities, armed conflicts, fragile health systems, and suboptimal community engagement, as well as political and economic instability, are considered as main reasons for the slow recovery of routine immunization in the WHO African Region [23,24]. The reemergence of vaccine-preventable disease outbreaks reported in several countries in the region, and particularly the ongoing diphtheria outbreak in West Africa, serve as a reminder of the increasing threat of infectious diseases due to low vaccination rates [25]. However, the WHO African Region recorded an increase in immunization coverage in 2022 compared to the pre-pandemic period with a few antigens such as the HEPBB and MCV2, despite remaining far below set global targets. This was mainly due to the fact that some countries introduced these vaccines during the pandemic period: 3 countries out of 15 for the HEPBB and 10 countries out of 40 for the MCV2.

The dramatic increase in the number of zero-dose children is one of the consequences of routine vaccination disruptions that led to low vaccination coverage. In this study, the WHO African Region recorded over 28 million zero-dose children from the pre-pandemic period (2019) to 2022, representing 19% of the target population from the last four years (2019–2022). It is well known that a high proportion of zero-dose children leads to gaps in population immunity and heightens the risk of child death and disease outbreaks [26,27]. One of the challenges of programs aiming at reducing the burden of zero-dose children is to identify missed communities for targeted and tailored interventions. Many zero-dose and under-vaccinated children live in challenging settings including remote rural areas, built-up and resource-poor urban settlements and areas experiencing conflicts and crises [28]. Hogan et al. [26] stated that reaching zero-dose children is key to achieving sustainable development goals (SDGs). These children face multiple deprivations related to education, water and sanitation, nutrition and access to other health services and account for one-third of all child deaths in low- and middle-income countries [26].

To this end, the “Big Catch-up” initiative [29], an essential immunization recovery plan for 2023 and beyond, represents a unique opportunity for catching up children who have missed immunization, restoring immunization services to the pre-pandemic levels and strengthening these services to achieve the targets of the Immunization Agenda 2030. However, reaching zero-dose children requires context-specific interventions to overcome barriers to vaccination that are multifaceted and nuanced to each setting [30]. In addition, mechanisms need to be in place to reduce drop-out in an equitable manner so that children are not only reached once but receive all the vaccines they need [31]. The quest to ensure that no child is left behind requires a tailored approach that addresses multiple and intersecting economic vulnerabilities, sociocultural barrier, and health system challenges to deliver immunization services through the primary healthcare system [32].

From 2019 to 2022, there were four cohorts of zero-dose and under-immunized children who, in 2023, are aged 12 to 59 months. Catching up vaccination for all these zero-dose cohorts may require adjusting immunization policies and schedules with national immunization technical advisory groups’ guidance [32] to remove restrictive target age groups or upper age limits for the Expanded Programmes on Immunization.

As part of such a process, local disease epidemiology, current immunization coverage levels and program performance, health system capacity, implications for budget and logistics should be taken into consideration, as recommended by the WHO [33]. The implementation of Big Catch-Up plans will result in increasing financing challenges to immunization programs due to constrained or shrinking health budgets [34]. It is critical for governments in the African Region to ensure that processes are in place for prioritizing immunization program investments, including for catch-up activities [34]. In February 2023, African Union Heads of State committed, through the declaration “Building Momentum for Routine Immunization Recovery in Africa”, to prioritizing universal access to immunization, increasing and sustaining domestic investments in vaccines as well as addressing bottlenecks in vaccine delivery and improving disease surveillance, with the shared goal of stopping and reversing the decline in immunization for zero-dose children [35]. In addition, in April 2023, Médecins Sans Frontières (MSF) called on Gavi, the Vaccine Alliance and other donors to expand the vaccine supply to ensure that all children up to the age of five are given the opportunity to catch up on their vaccinations.

In 2022, the administrative coverage of the DTP3 was greater than estimates from the WHO and UNICEF in 19 countries out of 47 (40%). This means that the WUENIC process downgraded the administrative coverage, highlighting data quality issues experienced by several countries in the WHO African Region [36]. Mihigo et al. [37] identified over-reporting and the underestimation of target populations as the main reasons for the over-estimation of immunization coverage in Nigeria. In most countries, target populations are estimated based on projections using old and inaccurate population census data, without applying WHO recommendations for methods for assessing target population accuracy, such as comparing estimates with alternative sources, plotting and analyzing target populations over time and monitoring target population growth rates [38]. Assessing the immunization data quality of routine reports in the Ho municipality of the Volta region in Ghana, Ziema and Asem [39] found data overreporting of 20% on children vaccinated with the BCG, DTP3 and MCV2. Reasons attributable to overreporting could be arithmetic errors during monthly data compilation or deliberate overreporting to achieve high coverage to avoid queries by higher staff levels [39]. Data quality issues leading to the over- or under-estimation of immunization coverage highlights the need to revamp the whole immunization information system.

## 5. Limitations

The WUENIC estimates are only made at the national level and cannot be used to guide operational decisions at sub-national levels. The latest available data on population estimates from the United Nations and WUENIC coverages are used to derive an estimation of the number of children who have missed all or part of the antigen for a scheduled immunization. This may result in the over- or under-estimation of the number of zero-dose and other under-immunized children [40]. The WUENIC estimates are made on a yearly basis and may inform timely decisions aimed at improving the immunization systems’ performance. The immunization coverage for some countries may have already improved or even deteriorated when the WUENIC estimates are published. The drivers of success or failure in achieving set global immunization targets were not analyzed as the WHO and UNICEF estimates do not include any qualitative information about countries’ vaccination programs.

The findings of this report should therefore be interpreted with these limitations.

## 6. Conclusions

The results of this study show that the WHO African Region has still not recovered from COVID-19’s disruptions to immunization services. In many countries in the region, immunization coverage for most lifesaving vaccines is still below 2019 levels, contributing to an increased risk of vaccine-preventable disease outbreaks.

As a result of low vaccination coverage during the emergency phase of the COVID-19 pandemic, millions of children partially or fully missed routine vaccines and have seen their vulnerability to vaccine-preventable diseases and the risk of death heightened. It is critical for governments in the African Region to restore immunization services and catch up with vaccination for un- and under-immunized children as a priority, in line with commitments made by African Union Heads of State in February 2023 toward building momentum for routine immunization recovery in Africa and, above all, to strengthen the vaccination system for greater resilience. All technical and financial partners are called on to intensify their support of countries’ efforts to mitigate the impact of COVID-19 on health systems in general and on immunization services, as well as to reduce the burden of zero-dose children, in particular.

The discrepancies between WHO and UNICEF estimates and administrative data on immunization coverage highlight the need to invest more in data quality for routine immunization.

## Figures and Tables

**Figure 1 vaccines-12-00168-f001:**
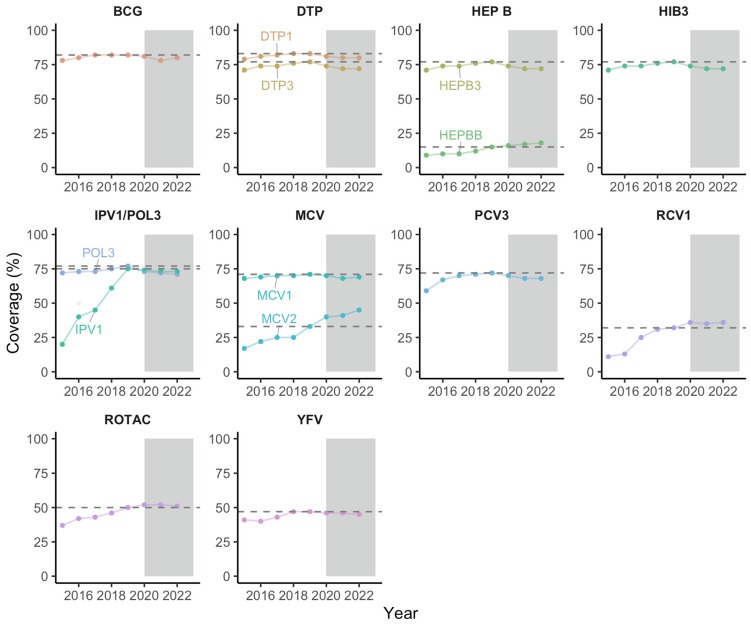
Coverage of the BCG, DTP1, DTP3, HEPB3, HEPBB, HIB3, IPV1, POL3, MCV1, MCV2, PCV3, RCV1, ROTAC and YFV over time in the African Region. The horizontal dashed lines highlight the coverage of each antigen or combination of antigens in 2019 to allow for a comparison with 2022 coverage. Abbreviations: BCG: Bacille Calmette–Guérin vaccine; DTP1: first dose of diphtheria–tetanus–pertussis-containing vaccine; DTP3: third dose of diphtheria–tetanus–pertussis-containing vaccine; HEPBB: hepatitis B vaccine, birth dose; HEPB3: third dose of hepatitis B vaccine; HIB3: third dose of Haemophilus influenzae type b vaccine; POL3: third dose of oral poliovirus vaccine; IPV1: first dose of inactivated-poliovirus-containing vaccine; MCV1: first dose of measles-containing vaccine; MCV2: second dose of measles-containing vaccine; PCV3: third dose of pneumococcal conjugate vaccine; RCV1: first dose of rubella-containing vaccine; ROTAC: final dose of rotavirus vaccine; YFV: yellow fever vaccine.

**Figure 2 vaccines-12-00168-f002:**
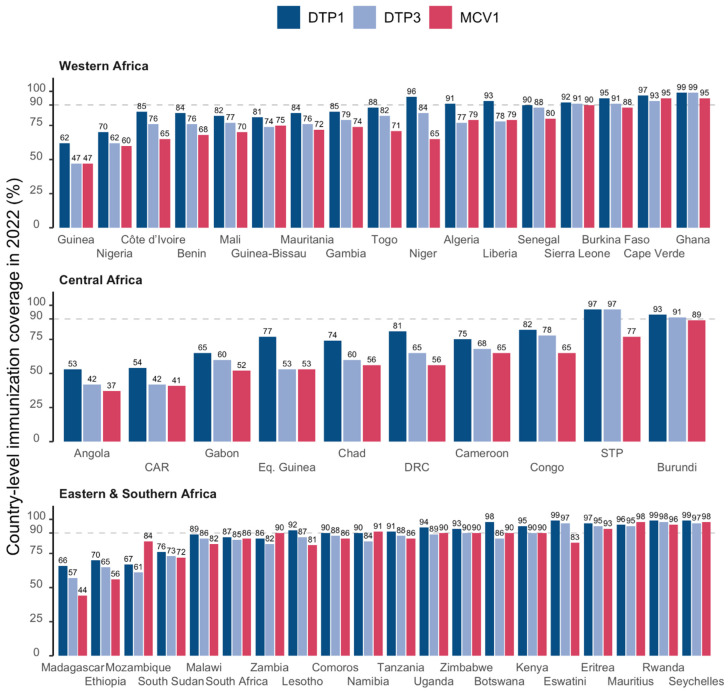
DTP1, DTP3 and MCV1 coverage by country in 2022 in the WHO African Region. The horizontal dashed lines represent the target coverage of 90%. Abbreviations: DTP1: first dose of diphtheria–tetanus–pertussis-containing vaccine; DTP3: third dose of diphtheria–tetanus–pertussis-containing vaccine; MCV1: first dose of measles-containing vaccine; CAR: Central African Republic; DRC: Democratic Republic of Congo; Eq. Guinea: Equatorial Guinea; STP: Sao Tome and Principe.

**Figure 3 vaccines-12-00168-f003:**
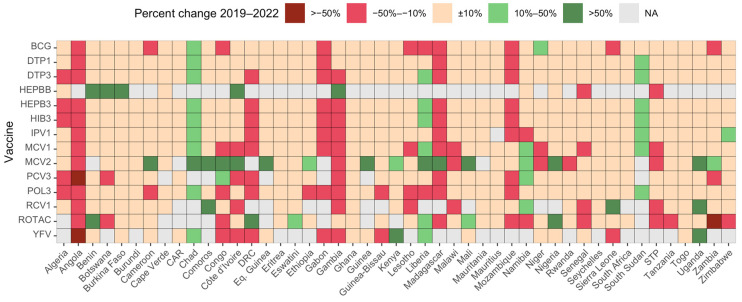
Percent change in immunization compared to 2019 in the 47 countries of the WHO African Region. Abbreviations: CAR: Central African Republic; DRC: Democratic Republic of Congo; Eq. Guinea: Equatorial Guinea; STP: Sao Tome and Principe; NA: data not available. BCG: Bacille Calmette–Guérin vaccine; DTP1: first dose of diphtheria–tetanus–pertussis-containing vaccine; DTP3: third dose of diphtheria–tetanus–pertussis-containing vaccine; HEPBB: hepatitis B vaccine birth dose; HEPB3: third dose of hepatitis B vaccine; HIB3: third dose of Haemophilus influenzae type b vaccine; POL3: third dose of oral poliovirus vaccine; IPV1: first dose of inactivated-poliovirus-containing vaccine; MCV1: first dose of measles-containing vaccine; MCV2: second dose of measles-containing vaccine; PCV3: third dose of pneumococcal conjugate vaccine; RCV1: first dose of rubella-containing vaccine; ROTAC: final dose of rotavirus vaccine; YFV: yellow fever vaccine.

**Figure 4 vaccines-12-00168-f004:**
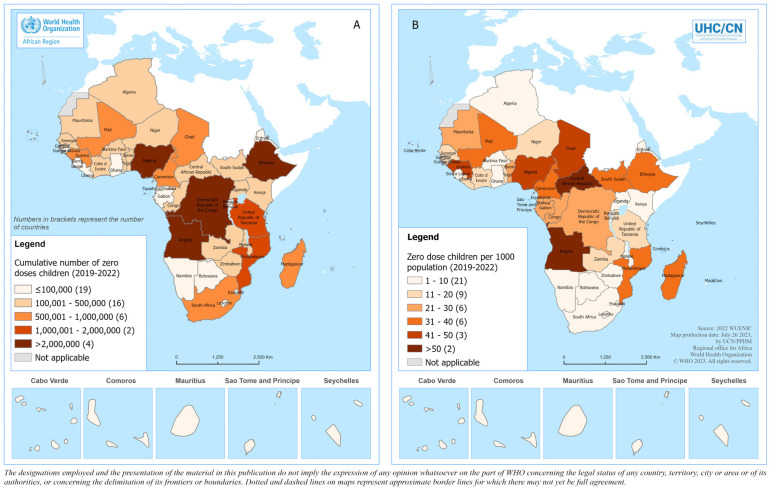
Geographical distribution of the (**A**) cumulative number of zero-dose children and (**B**) the cumulative number of zero-dose children per 1000 inhabitants in the WHO African Region during the pandemic period. The 2022 United Nations population estimates were used as the denominator [15].

**Figure 5 vaccines-12-00168-f005:**
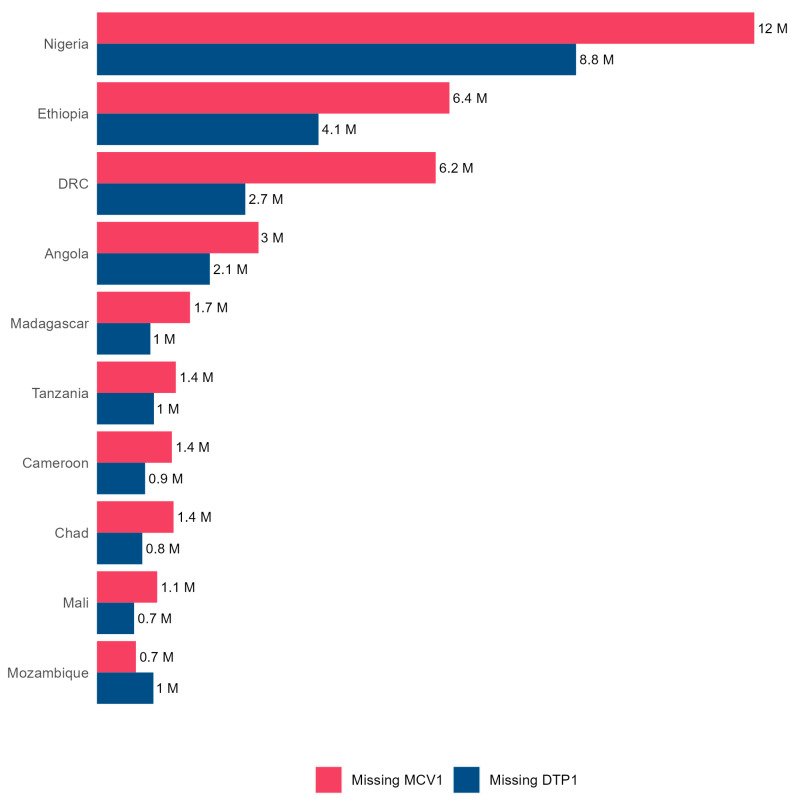
Cumulative number of unvaccinated children from 2019 to 2022 for the diphtheria–tetanus–pertussis (DTP)-containing vaccine and measles-containing vaccine (MCV) in the 10 countries of the WHO African Region with the highest burden of unvaccinated children.

**Figure 6 vaccines-12-00168-f006:**
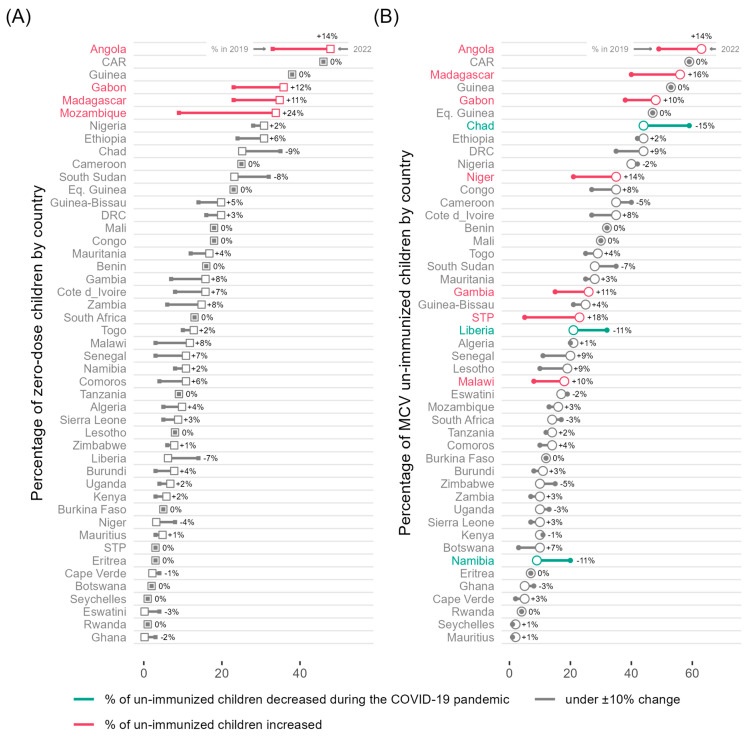
(**A**) The percentage of zero-dose children (i.e., children not immunized with the first dose of the diphtheria–tetanus–pertussis-containing vaccine (DTP1) in 2019 and 2022 and (**B**) the percentage of children not immunized with the first dose of the measles-containing vaccine (MCV1) in 2019 and 2022 by country in the WHO African Region. Abbreviations: CAR: Central African Republic; DRC: Democratic Republic of Congo; Eq. Guinea: Equatorial Guinea; STP: Sao Tome and Principe.

**Figure 7 vaccines-12-00168-f007:**
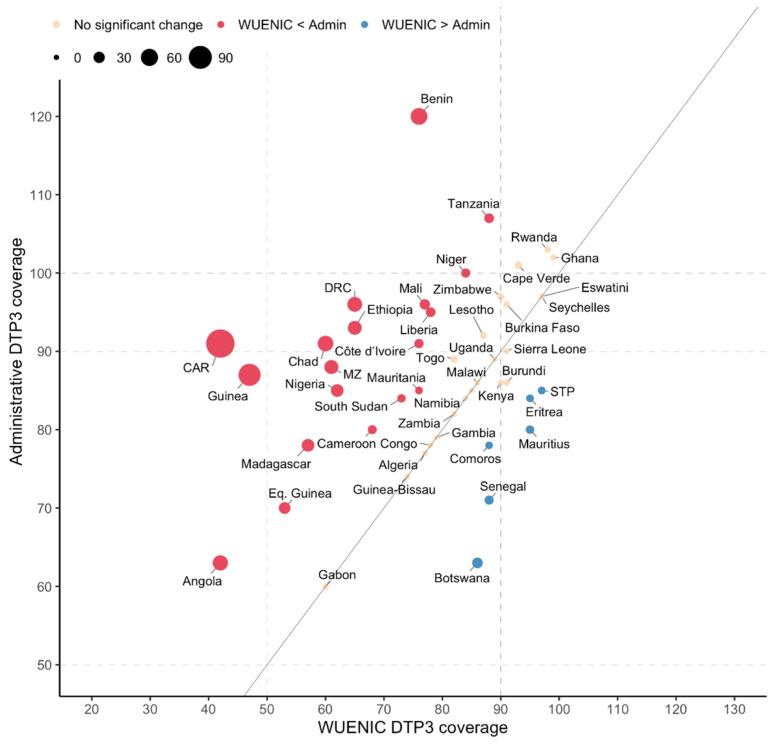
Percentage of change in the third dose of the diphtheria–tetanus–pertussis-containing vaccine (DTP3) coverage between the WUENIC and administrative reports in 2022 in 47 countries in the WHO African Region. Abbreviations: CAR: Central African Republic; DRC: Democratic Republic of Congo; STP: Sao Tome and Principe; Eq. Guinea: Equatorial Guinea; MZ: Mozambique.

**Table 1 vaccines-12-00168-t001:** Estimated number of zero-dose children in the WHO African Region from 2019 to 2022.

Year	# Surviving Children	Estimated Number of Vaccinated with DTP1	Estimated Number of Zero-Dose Children	% Zero-Dose Children
2019	36,995,277	30,763,363	6,231,914	16.8
2020	37,521,132	30,463,727	7,057,405	18.8
2021	38,080,516	30,439,539	7,640,977	20.1
2022	38,567,250	30,791,574	7,775,676	20.2
Cumulative 2019–2022	151,164,175	122,458,203	28,705,972	19.0

## Data Availability

The data presented in this study are available from the corresponding author upon request.

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
