# Peer review of "Status of Routine Immunization Coverage in the World Health Organization African Region Three Years into the COVID-19 Pandemic"

_vaccines, 2024, doi:10.3390/vaccines12020168_

Round 1
Reviewer 1 Report
Comments and Suggestions for Authors
Thank you for this very interesting article. It is timely and shows the important work that needs to be done to get children up to date on immunizations.
I have a few suggestions:
Figure 1- all the immunizations seem to follow the same pattern as the graphs except for HIB3. I would suggest you place the HIB3 abbreviation after the HEPB3 spot to be in line with the graphs. Currently, it is after the MCV2 spot not in keeping with the rest of the abbreviations.
Figure 3 is very difficult to read- is it needed in the manuscript or could it be in an appendix? It has valuable information but it is very difficult to read.
Limitations were discussed , I am happy to see you discussed the possible errors due to over/under reporting. A common problem with retrospective studies.
References: Some reference do not have the correct abbreviation but are spelled out fully.
#18- are no page numbers cited- is this a textbook reference?
I very much enjoyed reading your research it will bring to light an ongoing issue regarding immunizations.
Author Response
Reviewer: 1
- Comments to the Author
Thank you for this very interesting article. It is timely and shows the important work that needs to be done to get children up to date on immunizations.
Response: Thank you for this positive feedback.
Other comments:
- Figure 1- all the immunizations seem to follow the same pattern as the graphs except for HIB3. I would suggest you place the HIB3 abbreviation after the HEPB3 spot to be in line with the graphs. Currently, it is after the MCV2 spot not in keeping with the rest of the abbreviations.
Response: Thank you for this suggestion. We have placed the HIB3 abbreviation in the caption of Figure 1 and 3 in the same order in which they appear in the figures.
- Figure 3 is very difficult to read- is it needed in the manuscript or could it be in an appendix? It has valuable information but it is very difficult to read.
Response: Thank you for this comment. We have improved Figure 3 by adding black lines to the squares’ limits for better view and by increasing the size of x-axis text (country names). We believe that this figure is relevant to remain in the body of the manuscript.
- Limitations were discussed, I am happy to see you discussed the possible errors due to over/under reporting. A common problem with retrospective studies.
Response: Thank you for this suggestion. Possible under and over-estimation of administrative coverage was already highlighted in the discussion section (see lines 392-408”).
References: Some reference do not have the correct abbreviation but are spelled out fully.
Response: Thank you for noticing this formatting error, we have checked all the references and ensure that they all comply with the Journal’s guidance.
- #18- are no page numbers cited- is this a textbook reference?
Response: Thanks for this comment. We followed the citation format as instructed by the manual’s authors.
- I very much enjoyed reading your research it will bring to light an ongoing issue regarding immunizations.
Response: We thank you for this kind appreciation.
Reviewer 2 Report
Comments and Suggestions for Authors
This is a very well written manuscript that reports some very important results. The work of SAGE and the RITAGs post-pandemic is vital, given the disruptions to national and region immunisation programmes. The problems highlighted with immunisation coverage in this manuscript needs to drive regional immunisation policy makers.
The manuscript has some points that require some clarifications, and I have made some suggestions for each section separately. No major work is required, each comment is minor in scope and just seeks to clarify some things that the authors have already said.
**INTRODUCTION**
The research question has been stated but some clarifications are needed in this section:
Line 36: Please list the 7 African countries that are not included in the AFRO region, otherwise the reader will be confused.
Lines 49 to 52: The sentence beginning with “Zero-dose children refers…” can be split up and added appropriately in parentheses to the previous sentence. Line 48 should read “...resulting in a greater number of zero-dose (children who have not received any routine vaccination and more specifically …) and under-immunized children (those children missing the third dose of the DTP vaccine).”
Line 53: The reader might not be familiar with the IA2030 project so it needs to be briefly explained in this section before proceeding further.
**METHODS**
There is one main issue with the Methods section, which is that the two data sources have not been clearly defined and it was unclear after a first read that these two data sources would give such different results. The authors should properly define the Administrative coverage data source and the WUENIC data source, because the current definitions given in lines 80 to 86 are not clear enough. Perhaps a table showing the main attributes would help distinguish the two, but the authors are at liberty to work on this how they see fit.
Line 72: The pre-pandemic period here is defined as 2019, but on line 101 it states that data were taken from as far back as 2015. The authors need to clarify this point.
**RESULTS**
The authors have done a good job in presenting the numerous interesting and important results but there are some points that need to be strengthened.
Lines 129 to 130: The fact that all countries had data available for BCG, DTP1, etc. is impressive and really demonstrates the thorough work done by the authors with such huge datasets. What is puzzling is why fewer countries had data available for the MCV2, PCV3 etc. This needs to be discussed in the Discussion section, as there are no doubt various and important reasons for this systemic lack of data for these other vaccines.
Figure 1: The horizontal dashed lines need to be explained in the figure caption.
Figure 3: This is a brilliant method of presenting the data for each country, but I would prefer to see the +/-10% box divided into two separate boxes, one for to +10% and one for -10%. Merging the two means that we cannot tell if a country has reported a small increase in immunisation coverage or a small decrease. Although the largest decreases are the major causes for concern for policy makers, the smaller decreases are also important to notice, but this figure doesn’t allow that. Separately the Discussion section needs to go into some of these more, particularly the countries of Chad, Liberia, South Sudan, Uganda, Nigeria and Namibia.
Figure 4: This image is another example of an excellent method of presenting data to give a clear result. However, it would also be helpful to have the same data presented somewhere else, even in an appendix, as per-capita zero-dose children in AFRO. It is an important finding, although not completely surprising, that the countries with the highest numbers of zero-dose children are also some of the countries with the largest populations.
**DISCUSSION AND CONCLUSION**
The Discussion requires lots of additional discussion concerning the heterogeneous results presented in the previous section, and it requires some formatting to break-up the large dense paragraphs.
The authors need to add some discussion on the following results:
-
Figure 3, for the countries listed in the previous section
-
Figure 3, for the vaccines HEPBB, MCV2, PCV3, RCV1, ROTAR, YFV, because the lack of data for these vaccines is likely systemic and the reader needs to be made aware of why this is the case
-
Explain the results in Figure 7 relating to those countries that report >100% for the Administrative coverage and why this is the case. There’s a little bit of this in the paragraph that begins in line 313 but the authors could do more to highlight specific countries and explain the situation there
Line 261: Start a new paragraph at the beginning of the sentence “Conflicting public health…”
Line 274: Start a new paragraph at the beginning of the sentence “One of the challenges…”
Line 281: Start a new paragraph at the beginning of the sentence “To this end…”
Line 292: Start a new paragraph at the beginning of the sentence “From 2019 to 2022…”
Line 297: Start a new paragraph at the beginning of the sentence “As part of…”
The Conclusions section requires only a new paragraph starting on line 358 (“As a result of…”), on line 368 (“The discrepancies between…”).
**TITLE AND ABSTRACT**
The title accurately reflects the work presented in the body of the manuscript.
The abstract summarises well the main text of the manuscript. I have no suggestions to make for the abstract.
Author Response
Reviewer: 2 Comments to the Author
- This is a very well written manuscript that reports some very important results. The work of SAGE and the RITAGs post-pandemic is vital, given the disruptions to national and region immunisation programmes. The problems highlighted with immunisation coverage in this manuscript needs to drive regional immunisation policy makers. The manuscript has some points that require some clarifications, and I have made some suggestions for each section separately. No major work is required, each comment is minor in scope and just seeks to clarify some things that the authors have already said.
Response: We thank you for your kind review.
- INTRODUCTION
The research question has been stated but some clarifications are needed in this section: Line 36: Please list the 7 African countries that are not included in the AFRO region, otherwise the reader will be confused.
Response: Thanks for this suggestion. We have added the list of the African countries that are not part of the WHO African region (see lines 89-90).
- Lines 49 to 52: The sentence beginning with “Zero-dose children refers...” can be split up and added appropriately in parentheses to the previous sentence. Line 48 should read “...resulting in a greater number of zero-dose (children who have not received any routine vaccination and more specifically ...) and under-immunized children (those children missing the third dose of the DTP vaccine).”
Response: We thank you for the suggestion. We have modified been modified accordingly (lines 49-53).
- Line 53: The reader might not be familiar with the IA2030 project so it needs to be briefly explained in this section before proceeding further.
Response: We thank you for the suggestion, we have briefly described the IA2030(see lines 67-70).
- METHODS
There is one main issue with the Methods section, which is that the two data sources have not been clearly defined and it was unclear after a first read that these two data sources would give such different results. The authors should properly define the Administrative coverage data source and the WUENIC data source, because the current definitions given in lines 80 to 86 are not clear enough. Perhaps a table showing the main attributes would help distinguish the two, but the authors are at liberty to work on this how they see fit.
Response: Thanks for this comment/suggestion. We have clarified the difference between WUENIC and administrative data in the methods section (lines 94-97).
- Line 72: The pre-pandemic period here is defined as 2019, but on line 101 it states that data were taken from as far back as 2015. The authors need to clarify this point.
Response: The sentence has been modified to clarify that we analyzed data from 2015 to 2022 and compared the 2022 coverage to the pre-pandemic one (see lines 120-121).
- RESULTS
The authors have done a good job in presenting the numerous interesting and important results but there are some points that need to be strengthened. Lines 129 to 130: The fact that all countries had data available for BCG, DTP1, etc. is impressive and really demonstrates the thorough work done by the authors with such huge datasets. What is puzzling is why fewer countries had data available for the MCV2, PCV3 etc. This needs to be discussed in the Discussion section, as there are no doubt various and important reasons for this systemic lack of data for these other vaccines.
Response: We thank you for your comment. We have added the following sentence: “Countries that did report on these antigens (MCV2, ROTAC, RCV1, YFV and HEPBB) are those that have not yet introduced them in their national schedules” (see lines 155-156).
The issue is not related to the availability of data but to introduction of the said vaccines in national schedule. We did not discuss this as we are working on a separate paper related to factors associated with new vaccines introduction in the African region.
- Figure 1: The horizontal dashed lines need to be explained in the figure caption.
Response: Thank you for this suggestion. We have added a sentence that provides the meaning of dashed lines in the caption of Figure 1 and for Figure 2 as well.
- Figure 3: This is a brilliant method of presenting the data for each country, but I would prefer to see the +/-10% box divided into two separate boxes, one for to +10% and one for -10%. Merging the two means that we cannot tell if a country has reported a small increase in immunisation coverage or a small decrease. Although the largest decreases are the major causes for concern for policy makers, the smaller decreases are also important to notice, but this figure doesn’t allow that.
Response: We thank you for this suggestion. Given that the global target for most vaccines is 90%, and increase from 91% to 100% and a decrease from 90% to 81% represent +/-10% change. We considered non-significant, any change by <=+10% or >=-10%, and <=+10% or >=-10%, and did not found it relevant to add smaller class of changes. Moreover, adding additional classes of change would make the graph more difficult to read. Accordingly, we kept the same classes of change.
- Separately the Discussion section needs to go into some of these more, particularly the countries of Chad, Liberia, South Sudan, Uganda, Nigeria and Namibia.
Response: Thank you for this suggestion. We have provided additional information on antigens for which the highest increase was recorded as well as on countries that recorded the highest number of antigens with over +10% of change in coverage in 2022 compared to 2019 (See lines 214-227). The change in immunization coverage with HEPBB and MCV2 has been discussed (see lines 339-343). Given that the coverage for the three main antigens/combination of antigens (DTP1, DTP3 and MCV2) remained low in 2022 despite an increased compared to the 2019 level in Liberia, Chad, and South Sudan, we did not find it relevant to discuss such increase in coverage.
- Figure 4: This image is another example of an excellent method of presenting data to give a clear result. However, it would also be helpful to have the same data presented somewhere else, even in an appendix, as per-capita zero-dose children in AFRO. It is an important finding, although not completely surprising, that the countries with the highest numbers of zero-dose children are also some of the countries with the largest populations.
Response: We thank you for your comment. We have added a graph on the of number of zero-dose children per 1,000 population as part of Figure 4.
- DISCUSSION AND CONCLUSION
The Discussion requires lots of additional discussion concerning the heterogeneous results presented and in the previous section, and it requires some formatting to break-up the large dense paragraphs. The authors need to add some discussion on the following results:
- Figure 3, for the countries listed in the previous section
Response: Thanks for the suggestion. The change in immunization coverage with HEPBB and MCV2 has been discussed (see lines 339-343). Given that the coverage for the three main antigens/combination of antigens (DTP1, DTP3 and MCV2) remained low in 2022 despite an increase compared to the 2019 level in Liberia, Chad and South Sudan, we did not find it relevant to discuss such increase in coverage.
- 3. Figure 3, for the vaccines HEPBB, MCV2, PCV3, RCV1, ROTAR, YFV, because the lack of data for these vaccines is likely systemic and the reader needs to be made aware of why this is the case
Response: Thanks for the suggestion. Lack of these data in some countries is due to the fact that the related vaccine is not yet introduced in the national schedule and not to lack of data. This has been specified in the result section (see lines 155-156).
- Explain the results in Figure 7 relating to those countries that report >100% for the Ad- ministrative coverage and why this is the case. There’s a little bit of this in the paragraph that begins in line 313 but the authors could do more to highlight specific countries and explain the situation there
Response: Thanks for the suggestion. Countries with over 100% of administrative coverage have been listed (see lines 308-311). The overall issue of data quality leading to over or under-estimation of coverage has been discussed (see lines 392-407) without pointing out specific countries as this is known to be due to overreporting and over/under-estimation of target population.
- Line 261: Start a new paragraph at the beginning of the sentence “Conflicting public health...” Line 274: Start a new paragraph at the beginning of the sentence “One of the challenges...” Line 281: Start a new paragraph at the beginning of the sentence “To this end...” Line 292: Start a new paragraph at the beginning of the sentence “From 2019 to 2022...” Line 297: Start a new paragraph at the beginning of the sentence “As part of...” The Conclusions section requires only a new paragraph starting on line 358 (“As a result of...”), on line 368 (“The discrepancies between...”).
Response: Thanks for the suggestion. We have made the suggested changes.
- **TITLE AND ABSTRACT** The title accurately reflects the work presented in the body of the manuscript. The abstract summarises well the main text of the manuscript. I have no suggestions to make for the abstract.
Response: We thank you for your kind review.
